# The Latency Performance Analysis and Effective Relay Selection for Visible Light Networks

**DOI:** 10.3390/s24092748

**Published:** 2024-04-25

**Authors:** Baozhu Yu, Xiangyu Liu

**Affiliations:** School of Information Science and Engineering, Shenyang Ligong University, Shenyang 110158, China; yubz19@mails.jlu.edu.cn

**Keywords:** latency, statistical reliability, martingales, aggregate traffic, forwarding scheme

## Abstract

In visible light communication (VLC), the precise latency evaluation of wireless access networks and the efficient forwarding strategy of core networks are the crux for end-to-end reliability provisioning. Leveraging martingale theory, an elegant latency-bounded reliability analysis framework is studied for the VLC network. Considering the characteristic that VLC links are easy to block, the martingale of the service process is constructed. Based on the time shift features, the martingale process related to latency is proposed for the VLC system, which models the influence of entanglement between aggregate arrivals and random service on latency. A stopping time event about latency is defined. Renting the stopping time theory, a tight upper bound of the unreliability with regard to latency is derived. In the core network, a dynamic forward backhaul framework is proposed, which uses the relay selection algorithm based on back-pressure theory to improve data transmission quality. The theoretical latency-bounded reliability matches the simulation results well, which verifies the effectiveness of the proposed analysis framework, and the proposed relay selection algorithm can also improve network performance under data-intensive transmission.

## 1. Introduction

In 5G and beyond, massive businesses with strict reliability requirements surge, which expects extensive bandwidth provisioning. This will lead to more severe spectrum shortages. Visible light communication (VLC) is a potential paradigm to ease the dilemma of radio frequency (RF) resources in the next-generation communication systems [1,2]. Benefiting from the characteristics of light, VLC can provide high transmission rates, ultra-wide frequency bands, high-energy efficiency, low cost, and indoor full coverage. It can be deployed as indoor high-speed data links for personal area networks and RFs in non-friendly environments. In critical scenarios, such as remote control in medical operating rooms and factory automation under strong magnetic interference, VLC is expected to embrace massive services with traffic possessing strong randomness, heterogeneity, and burstiness [3]. The data require the reliability of latency within the millisecond level to reach 99.99–99.999% [4,5]. For traffic featuring complex characteristics transmitted on visible light channels, we should explore an effective and precise evaluation method of reliability with regard to latency, which can reveal the influence of entanglement between visible light data traffic and visible light transmission schemes. Meanwhile, an efficient routing strategy needs to be explored for the VLC network.

In the VLC wireless network, the light-emitting diode (LED) access point (AP) carries packet flows from different services, which possess heterogeneous characteristics [6,7]. The convergence of heterogeneous visible light data flows intensifies the uncertainty of the aggregate traffic. In [8], the aggregate traffic was composed of multiple 2-state Markov-modulated Bernoulli processes (MMBP-2) and was tackled as one MMBP-2 by the Kronecker product operation. The proposed method is not of universal applicability and is unable to provide a theoretical result of the latency-bounded reliability. The stochastic network calculus (SNC) theory, effective bandwidth (EB), and effective capacity (EC) theories provide classical analysis methods of latency-bounded reliability. The union-bound inequality, as the key enabler in SNC, only supports capturing a loose upper bound of unreliability [9]. In the EB/EC framework, the stochastic traffic is tackled as a constant flow [10,11], which always triggers rougher theoretical results of latency performance, especially for bursty traffic.

Martingale, a random process throughout modern probability theory, has demonstrated great superiority in the precise analysis of the statistical reliability regarding latency. The theoretical upper bounds of unreliability derived in [9,12] are more precise than the existing conclusions. This martingale-based analysis framework is a milestone. In [13,14], the latency-bounded unreliability of the Markov-modulated on–off (MMOO) arrival process was analyzed in the ALOHA access scheme. Based on the results, the bandwidth resources were allocated reasonably. More importantly, martingales have been explored to support the evaluation of the end-to-end reliability of latency. In [15], a precise analysis framework of the end-to-end latency-bounded unreliability was introduced, where a multi-hop routing path was considered and the traffic was modeled as a Markov-modulated process. Then, the proposed method was extended in the 6G network scenarios [16]. For the system adopting the THz wireless access scheme, the end-to-end latency performance was analyzed for the traffic generated from virtual reality. In [17], we constructed an analysis framework of latency performance for the aggregate traffic, which was constituted by the heterogeneous Markov-modulated flows. The martingale of the queuing length was defined, and the latency-bounded unreliability was derived.

These remarkable conclusions have inspired our work on the statistical latency QoS analysis based on martingale theory. The framework in [17] is theoretical without specific application scenarios. In this paper, the latency analysis in the VLC network is focused on. The differences with [17] can be summarized as follows. (1) The cross-layer service process is modeled, which is the improvement of the proposed method in [17]. Specifically, in the physical layer, the Lambertian sources are considered and the channel gain is depicted. The Shannon Theorem is leveraged to model the achievable transmission rates, which achieves the mapping between the physical layer and the data link layer. We focus on the characteristic that the VLC links are easy to block. Blocking the line of sight (LoS) between the VLC AP and the targeted terminal at each time slot is considered a random variable, which triggers the stochastic features of the service process and further impacts the latency of the aggregate traffic. (2) Meanwhile, the martingale process related to latency is a proposed novelty, which is another way to analyze latency different from [17]. It introduces the statistical characteristics of the service scheme and data traffic into martingale parameters, which reveals the influence of random blocks of the VLC link and the burstiness of aggregate traffic on the latency. (3) The stopping time event about latency is defined, which is the time point when the system latency exceeds the threshold. Applying the stopping time theory to the latency-related martingale, the upper bound of the unreliability with regard to latency is captured. (4) More importantly, the end-to-end reliability provisioning is investigated in this paper, not just the air interface latency analysis.

Further, for the aggregate traffic forwarded by the VLC AP, an efficient routing algorithm is proposed based on the back-pressure theory in the core network. Some studies have investigated the utilization of back-pressure theory to improve network performance, specifically within the Transmission Control Protocol domain [18]. Additionally, alternative research has concentrated on employing back-pressure algorithms to facilitate relay selection in wireless sensor networks and wireless multi-hop networks [19]. Although the source node (LED access point) can adopt the classic back-pressure algorithm to promote the network capacity, the classic algorithm results in significant packet delays and poor node energy. Therefore, combining the latency performance analysis, we design a new link weight calculation method to make routing and scheduling decisions, which considers the data packet’s recent node record, the neighbor node’s remaining energy status, and the queue latency. Meanwhile, we note that no prior research has explored the integration of back-pressure theory in visible light network systems.

To summarize, in this paper, considering the VLC APs serve the aggregate traffic with burstiness, a precise analysis framework of the latency-bounded reliability is proposed. In the core network, an efficient routing algorithm is introduced for the complex aggregate traffic. The contributions can be concluded as follows.

The service process of the VLC network and the aggregate arrival process are modeled in the martingale domain. For the VLC link with a random block, the i.i.d. service process provided by the LED AP is considered. Through mapping the statistical features of random block behavior in the martingale parameters, the martingale of the service is constructed. Leveraging the relation between the spectral radius and the eigenvector in Markov chains, martingales of Markov-modulated arrival processes are defined.We propose a martingale process related to latency, which models the impact of entanglement between complex arrivals and random service on the latency. A stopping time point at which the latency violates the threshold is considered. Based on the stopping time theory, the complementary cumulative distribution function of latency is obtained for the VLC network. We evaluate the latency in terms of burstiness, which is measured by the squared coefficient of variation.When the LED AP node data reach the core network, a dynamic back-pressure algorithm based on energy and latency from aggregate traffic is proposed, which designs a new link weight calculation method to make routing and scheduling decisions. The modified back-pressure algorithm trades off three factors, including the data packet’s recent node record, the neighbor node’s remaining energy status, and the queue latency, to boost the network performance and data transmission quality. Meanwhile, our scheme is the first research to apply back-pressure theory in the visible light network.

## 2. The Network Model and the Queuing System

We consider a multi-service uplink communication scenario, as shown in Figure 1. The data flow from different services (business) is statistically heterogeneous. That is, diverse models are adopted to describe the randomness of flows. The data packets belonging to different flows could access the same VLC AP, where these flows compose the aggregate traffic. The targeted AP supplies the transmission rates for the aggregate traffic according to a specific scheme. Because of the randomness of the VLC channel and the block, the transmission process provided for the aggregate traffic is stochastic. Thus, the backlog and latency are triggered in the buffer of the targeted VLC AP.

***A.*** 
*
**The queuing system of aggregate traffic**
*


In the VLC access network, the target terminal is associated with the nearest VLC AP. This terminal carries two services where the generated packet flows are heterogeneous. Thus, the aggregate traffic is transmitted from this terminal, as shown in Figure 1a. We consider the block of LoS between the VLC AP, and this terminal obeys a Bernoulli distribution of parameter 1−p. Then, the channel gain can be modeled as
(1)$$\begin{array}{c} hv=\left\{\begin{array}{c} \frac{(b+1)Ap}{2\pi d^{2}}\cos^{b}\left(\varphi^{ir}\right)g_{of}f\left(\varphi^{in}\right)\cos\left(\varphi^{in}\right),\;p\\ 0,1-p \end{array}\right., \end{array}$$
where *b* is the Lambertian index, and $b=-1/\log_{2}\left(\cos\left(\vartheta_{1/2}\right)\right)$. $\vartheta_{1/2}$ is the half-intensity radiation angle. Ap is the area of reviewer photodetector (PD). *d* is the distance between the AP and the terminal. $g_{of}$ is the gain of the optical filter. $\varphi^{in}$ and $\varphi^{ir}$ are the angle of incidence and irradiance between VLC AP and the terminal, respectively. $f(\varphi^{in})$ is the optical concentrator gain, which is a function of $\varphi^{in}$
(2)$$\begin{array}{c} f\left(\varphi^{in}\right)=\left\{\begin{array}{c} \frac{r^{2}}{\sin^{2}\left(\Theta \right)},\;0\leq \varphi^{in}\leq \Theta \\ 0,\;\varphi^{in}>\Theta \end{array}\right. \end{array}$$
where *r* is the refractive index, and Θ is the semi-angle of the field of view (FoV) of the PD.

Based on the Shannon theorem, the achievable transmission rate between AP and the terminal, which is defined as $R^{VLC}\left(n\right)$, can be given as
(3)$$\begin{array}{c} \begin{array}{c} R^{VLC}\left(n\right)=W_{VLC}\times \log_{2}\left(1+\frac{p\left(n\right)hv^{2}}{{\sigma_{VLC}}^{2}}\right)\left[bits/s\right], \end{array} \end{array}$$
where $W_{VLC}$ is the system bandwidth, and p(n) is the transmission power. ${\sigma_{VLC}}^{2}$ is the noise power.

The access process of the packets from aggregate traffic can be modeled as a queuing system, as is presented in Figure 1b. These two flows constituting the aggregate traffic are independent and described as the four-state Markov-modulated multinomial process (MMMP-4) and the interrupted multimomial process (IMP), respectively, which are introduced in part B in detail. These are the arrival processes of the queuing system. The access process is modeled as the service process provided by the AP. Because latency is often described in units of packets, the arrival process and the service process are modeled from the perspective of packets/slots. Without loss of generality, the length of packets is assumed to be fixed. The flows from different services have the same priority. Thus, the first-in-first-out (FIFO) scheduling scheme is adopted in the buffer.

In Figure 1b, $A_{i}\left(m,n\right),i=1,2$ is the accumulated arrival processes in [m,n] of flow *i*. $A_{i}\left(m,n\right)$ can be written as $A_{i}\left(m,n\right)=\sum_{k=m}^{n}a_{i}\left(k\right)$, where $a_{i}\left(k\right)$ is the number of arrival packets of flow *i* at time slot *k*. The arrival process $\left\{a_{1}\left(n\right),n\geq 0\right\}$ is modeled as an MMMP-4, and $\left\{a_{2}\left(n\right),n\geq 0\right\}$ is modeled as an IMP, which are bursty and heterogeneous. Let A(m,n) denote the aggregate accumulated arrival process from slot *m* to slot *n*. Similarly, the accumulated service process S(m,n) provided by the VLC AP is also a bivariate process, $S\left(m,n\right)=\sum_{k=m}^{n}s(k)$, where {s(n),n≥0} is the service process. s(n) is the number of the served packets at slot *n*. In this paper, s(n) is a random variable following a Bernoulli distribution with successful probability *p* and service rate *C*. s(n) can be given by
(4)$$s\left(n\right)=\left\{\begin{array}{c} C\;\;\;\;\;\;\;p\\ 0\;\;\;1-p \end{array}\right.\;\begin{array}{c} \mathrm{packets}/\mathrm{slot} \end{array},$$
where $C=\left\lfloor \frac{R^{VLC}\left(n\right)T}{L}\right\rfloor$. *T* s/slot is the duration of a slot and *L* bits/packet is the length of a packet. To simplify the subsequent analysis, *C* is assumed as a constant.

Based on SNC, the departure process D(0,n) can be defined by A(0,n) and S(0,n) using (min, +) convolution. The backlog process {Q(n),n≥0} in the buffer is defined as
(5)$$\begin{array}{c} Q\left(n\right)=\underset{n\geq 0}{\sup}\left\{A\left(0,\;n\right)-S\left(0,\;n\right)\right\}, \end{array}$$
and the latency W(n) at slot *n* is
(6)$$\begin{array}{c} \begin{array}{c} W(n)=\inf\{k\geq 0:A(0,n-k)\leq D(0,n)\}. \end{array} \end{array}$$

***B.*** 
*
**The heterogeneous arrival models**
*


The MMMP-4 is proposed to describe the interweaving arrival characteristics of packet flows from different services in a terminal. A 2D Markov chain is considered, which is denoted as $\left\{X_{1}\left(n\right),Z_{1}\left(n\right)\right\}$. $X_{1}\left(n\right)=\left\{U_{1},V_{1}\right\}$ represents the service that generates the packets. $Z_{1}\left(n\right)=\left\{N,Y\right\}$ models whether there are packets generated at time slot *n*. $Z_{1}\left(n\right)=N$ means that no packet is arriving, while $Z_{1}\left(n\right)=Y$ means that packets are generated. Corresponding to $U_{1}$ and $V_{1}$, packet arrival probabilities at each slot are $\lambda_{1}$ and $\gamma_{1}$, respectively. If service $U_{1}$ generates packets at slot *n*, the number of packets is $R_{M1}$, and in the other case, it is $R_{M2}$. Thus, the state space of the MMMP-4 model is $\left\{\left(U_{1},N\right),\left(U_{1},Y\right),\left(V_{1},N\right),\left(V_{1},Y\right)\right\}$. The state transition probabilities between two services, $U_{1}$ and $V_{1}$, are $p_{1}$ and $q_{1}$, respectively. Based on [17], the state transition matrix of the 2D Markov chain is defined as $T_{1}$.

The IMP model is proposed to embody the bursty and sporadic arrival characteristics of the small data service. It is also a 2D Markov chain, which is denoted as $\left\{X_{2}\left(n\right),Z_{2}\left(n\right)\right\}$. $X_{2}\left(n\right)=\left\{U_{2},V_{2}\right\}$. $V_{2}$ represents the service that generates the packets. In state $U_{2}$, there are no packets generated. $Z_{2}\left(n\right)=\left\{N,Y,I\right\}$ describes whether there are packets at slot *n*. $Z_{2}\left(n\right)=N$ represents that no packet arrives in state $V_{2}$, and $Z_{2}\left(n\right)=Y$ represents that $R_{I2}$ packets arrive in state $V_{2}$. It is worth noting that $Z_{2}\left(n\right)=I$ means that no packets arrive in idle state $U_{2}$ with probability one. The state space of the IMP model is $\left\{\left(U_{2},I\right),\left(V_{2},N\right),\left(V_{2},Y\right)\right\}$. The state transition probabilities between $U_{2}$ and $V_{2}$ are $p_{2}$ and $q_{2}$, respectively. In $V_{2}$, the packet generation process is stochastic and follows a multinomial distribution with the arrival probability $\gamma_{2}$ and the number of arrival packets $R_{I2}$. The state transition matrix of the IMP is $T_{2}$. The MMMP-4 model and IMP model are shown in [17].

## 3. Martingale-Based Latency-Bounded Reliability Analysis

The arrival and service processes are described by martingales in this part. The upper bound of the latency-bounded unreliability is analyzed based on the stopping theory.

***A.*** 
*
**Martingale constructions of arrival and service processes**
*


Firstly, we introduce the definition of martingales.

**Definition** **1** (martingales)**.**
*Let $\left(\Omega ,F_{\infty },P\right)$ be a probability space and ${\left\{F_{t}\right\}}_{t\geq 0}$ be a filtration, i.e., a non-decreasing sequence of σ-fields of $F_{\infty }$, $F_{t}\subset F_{t+1},t\geq 0$. A random sequence {X(t),t≥0} is adapted to ${\left\{F_{t}\right\}}_{t\geq 0}$. That is, X(t) is measurable with respect to $F_{t},\forall t\geq 0$. If*
*(1)* 

E[|X(t)|]<∞,∀t≥0

*(2)* 
*$\;E[X\left(t+1\right)|F_{t}]=X\left(t\right),\forall t\geq 0$,*

*holds, then {X(t),t≥0} is a martingale sequence.*


Further, if $\;E[X\left(t+1\right)|F_{t}]\leq X\left(t\right)$, then {X(t),t≥0} is a supermartingale.

As a powerful mathematical method, martingales can provide us with a framework to depict heterogeneous arrivals with burstiness. Martingale processes are inclusive, which shows that other random processes are transformed as martingale processes by martingale construction. Meanwhile, martingale can provide a modularized description method for arrival and service processes, which contributes to the model, analyzes the features of the backlog process (the difference between two random processes), and further enables the analysis of latency for aggregate traffic. It is noted that the inequality formulary of martingales supports the upper bound analysis of unreliability with regard to latency naturally. We construct the martingale processes of arrival and service, respectively, as follows.

**Definition** **2** (Martingale of arrival)**.**
*Consider a Markov process {x(n),n≥0}, which is $F_{t}$-measurable. The state space is N={1,2,⋯,N}. The state transition matrix is $T={\left[T_{i,j}\right]}_{N\times N}$. $T_{i,j}=P\left\{x\left(t+1\right)=j|x\left(t\right)=i\right\}$. The arrival process is Markov modulated, i.e., a(t)=f(x(t)) and $A\left(m,n\right)=\sum_{j=m}^{t}f(x(j)),n\geq m\geq 0$. Define a θ-transform for T as $T^{\theta }={\left[T_{i,j}^{\theta }\right]}_{N\times N},$∀θ>0. $T_{i,j}^{\theta }=T_{i,j}e^{\theta f(x(t+1))}$. The spectral radius of $T^{\theta }$ is $sp\left(T^{\theta }\right),sp\left(T^{\theta }\right)>0$, and the corresponding eigenvector is $V^{a}={\left[V_{i}^{a}\right]}_{N\times 1},V^{a}≻0$. The random process*

(7)
$$\begin{array}{c} M^{a}\left(m,n\right)=V_{x(n)}^{a}e^{\theta [A(m,n)-(n-m)Ka]}, \end{array}$$

*is constructed as a martingale relative to ${\left\{F_{t}\right\}}_{t\geq 0}$. Then, $M^{a}\left(m,n\right)$ is admitted as a martingale of arrival {a(n),n≥0}. The function Ka is*

(8)
$$\begin{array}{c} Ka=\frac{\log sp(T^{\theta })}{\theta }. \end{array}$$



We use the definition of martingales to prove Definition 2.

**Proof.** It is obvious that $sp(T^{\theta })<\infty$ and $V_{x(t)}^{a}<\infty$. Thus, $E[|M^{a}\left(m,n\right)|]<\infty ,\forall n\geq 0$ holds. We can write that
(9)$$\begin{array}{c} \begin{array}{c} E\left[V_{x(n+1)}^{a}e^{\theta [A(m,n+1)-(n+1-m)Ka]}|F_{n}\right]\\ =e^{\theta [A(m,n)-(n-m)Ka]}e^{-\theta Ka}\\ \;\times \sum_{x(n+1)\in N}V_{x(n+1)}^{a}e^{\theta f(x(n+1))}P\left\{x\left(n+1\right)|x\left(n\right)\right\}\\ \overset{\left(a\right)}{=}e^{\theta [A(m,n)-(n-m)Ka]}e^{-\theta Ka}\\ \;\times \sum_{x(n+1)\in N}V_{x(n+1)}^{a}T_{x(n),x(n+1)}^{\theta }\\ =e^{\theta [A(m,n)-(n-m)Ka]}e^{-\theta Ka}{\left(T^{\theta }V\right)}_{x(n)}^{a}\\ \overset{\left(b\right)}{=}e^{\theta [A(m,n)-(n-m)Ka]}e^{-\theta Ka}sp\left(T^{\theta }\right)V_{x(n)}^{a}\\ =M^{a}\left(m,n\right) \end{array} \end{array}$$
(a) is from the basic operation of conditional expectation and the definition of $T_{i,j}^{\theta }$. ${\left(T^{\theta }V^{a}\right)}_{i}$ is the *i*th element of vector $\left(T^{\theta }V^{a}\right)$. Then, (b) holds due to the relation between the eigenvector and eigenvalue, i.e., $T^{\theta }V^{a}=sp\left(T^{\theta }\right)V^{a}$, and the definition of Ka.    □

For the MMMP-4 and IMP models, the martingale parameters are $V_{x_{1}\left(t\right)}^{a}$, $Ka_{1}$, and $V_{x_{2}\left(t\right)}^{a}$, $Ka_{2}$ respectively, which are related to $T_{1}$ and $T_{2}$. The corresponding specific formulas can refer to [17].

Similar to the martingale of arrival, the martingale of the i.i.d. service process is constructed as described in Definition 3.

**Definition** **3** (Martingale of service)**.**

*The service process is {s(n),n≥0}, which is $G_{n}$-measurable. The moment generating function is $\Psi \left(\theta \right)=E[e^{\theta s(n)}]<\infty ,$∀θ>0. The accumulated service process of s(n) is {S(m,n),n≥m≥0}. Then, the random process*

(10)
$$\begin{array}{c} M^{s}\left(m,n\right)=V_{s(n)}^{s}e^{\theta [(n-m)Ks-S(m,n)]}, \end{array}$$

*is constructed as a martingale relative to ${\left\{G_{n}\right\}}_{n\geq 0}$. $M^{s}\left(m,n\right)$ is admitted as a martingale of service {s(n),n≥0}. The function Ks is*

(11)
$$\begin{array}{c} Ks=-\frac{\Psi (-\theta )}{\theta }. \end{array}$$



**Proof.** According to Ψ(θ)<∞,∀θ>0, we can get $E[|M^{s}\left(m,n\right)|]<\infty ,\forall n\geq 0$. Meanwhile,
(12)$$\begin{array}{c} \begin{array}{c} E\left[V_{s(n)}^{s}e^{\theta [(n+1-m)Ks-S(m,n+1)]}|G_{n}\right]\\ \overset{\left(a\right)}{=}e^{\theta [(n-m)Ks-S(m,n)]}e^{\theta Ks}E\left[V_{s(n+1)}^{s}e^{-\theta s(n+1)}\right]\\ \overset{\left(b\right)}{=}e^{\theta [(n-m)Ks-S(m,n)]}e^{\theta Ks}V_{s(n)}^{s}\left[e^{-\theta s(n+1)}\right]\\ =M^{s}\left(m,n\right). \end{array} \end{array}$$
(a) is because of the i.i.d. property. To support formula (b), $V_{s(n)}^{s}$ is set as a constant. The proof is completed based on $\Psi \left(-\theta \right)=E\left[e^{-\theta s(n+1)}\right]$.    □

Based on the characteristics of martingale processes, the backlog process can be modeled in the martingale domain, which facilitates the analysis of the latency performance.

**Definition** **4** (Martingale of backlog)**.**
*The martingales of arrival and service are defined as Definition 2 and 3. Define the product of (Equation 7) and (Equation 10) as the martingale of backlog $M^{L}\left(n\right)$ by the restriction of $\theta^{*}$,*

(13)
$$\begin{array}{c} \begin{array}{c} M^{L}\left(n\right)=\prod_{i=1}^{2}V_{x_{i}\left(n\right)}^{a}V_{s(n)}^{s}e^{\theta^{*}\left(A\left(m,\;n\right)-S\left(m,\;n\right)\right)}, \end{array} \end{array}$$

*where $\theta^{*}=sup\left\{\theta >0:Ka_{1}+Ka_{2}\leq Ks\right\}$.*


***B.*** 
*
**The latency-bounded unreliability**
*


**Theorem 1.** 
*Consider a link for the aggregate traffic. The service process is modeled as {s(n),n≥0}. The aggregate traffic is $\left\{a_{1}\left(n\right)+a_{2}\left(n\right),n\geq 0\right\}$. The martingale of arrival is defined by Definition 2, and the martingale of service is in Definition 3. Then, the latency-bounded unreliability of the link holds for ∀k>0,*

(14)
$$\begin{array}{c} P\left\{W\left(n\right)\geq k\right\}\leq \frac{E\left[V_{x_{1}\left(0\right)}^{a}\right]E\left[V_{x_{2}\left(0\right)}^{a}\right]E\left[V_{s(0)}^{s}\right]}{H}e^{-\theta^{*}Ksk}, \end{array}$$

*where $H=min\{V_{x_{1}\left(n\right)}^{a}V_{x_{1}\left(n\right)}^{a}V_{s(n)}^{s}:a_{1}\left(n\right)+a_{2}\left(n\right)-s\left(n\right)>0\}$. k is the latency threshold.*


The proof of Theorem 1 is based on the stopping time theory of martingale, as shown in Theorem 2 of [17].

**Proof.** Define a stopping time *T* for the martingale of backlog $M^{L}\left(n\right)$, which is the first time that the backlog exceeds threshold σ, i.e., T=min{n:A(0,n)−S(0,n)≥σ}. Let T∧n=min{T,n}. For $M^{L}\left(n\right)$, Theorem 2 is used. The probability of system backlog exceeding threshold σ can be obtained
(15)$$\begin{array}{c} P\left\{Q\left(n\right)\geq \sigma \right\}\leq \frac{E\left[V_{x_{1}\left(0\right)}^{a}\right]E\left[V_{x_{2}\left(0\right)}^{a}\right]E\left[V_{s(0)}^{s}\right]}{H}e^{-\theta^{*}\sigma }. \end{array}$$The detailed derivation process can refer to [17]. Based on the definition of the departure process, the latency-bounded probability, P{W(n)≥k} can be transformed as follows.
(16)$$\begin{array}{c} \begin{array}{c} P\left\{W\left(n\right)\geq k\right\}=P\left\{A(0,\;n-k)\geq D(0,\;n)\right\}\\ \leq P\left\{\max_{n-k\geq l}\left\{A\left(l,n-k\right)-S\left(l,\;n\right)\right\}\geq 0\right\}\\ =P\left\{\begin{array}{c} \max_{n-k\geq l}\{A_{1}\left(l,n-k\right)-\left(n-k-l\right)Ka_{1}+A_{2}\left(l,n-k\right)-\left(n-k-l\right)Ka_{2}\\ \;\;\;\;\;\;\;+(n-l)Ks-S(l,\;n)\}\geq kKs \end{array}\right\} \end{array} \end{array}$$
According to the time shift feature of martingales, it is obvious that
(17)$$\begin{array}{c} \begin{array}{c} M_{l}^{L-s}\left(n-k\right)=\prod_{i=1}^{2}V_{x_{i}\left(n-k\right)}^{a}V_{s(n-k)}^{s}e^{\theta^{*}\left(A\left(l,\;n-k\right)-S\left(l,\;n\right)\right)}, \end{array} \end{array}$$
is also a martingale process. $M_{l}^{L-s}\left(n-k\right)$ can be regarded as a martingale related to latency. To complete the latency analysis, the stopping time event about latency is defined as SW={W(n)≥k} and can be transformed as
(18)$$\begin{array}{c} SW=\{A_{1}\left(l,n-k\right)-\left(n-k-l\right)Ka_{1}+A_{2}\left(l,n-k\right)\\ -\left(n-k-l\right)Ka_{2}+\left(n-l\right)Ks-S\left(l,\;n\right)\geq kKs\} \end{array}$$
Then, the first time slot that SW occurs is the stopping time $T_{W}$.
(19)$$\begin{array}{cc} T_{W}=\inf\{n>0:A_{1}\left(l,n-k\right)-\left(n-k-l\right)Ka_{1} & +A_{2}\left(l,n-k\right)-\left(n-k-l\right)Ka_{2}\\ \; & +(n-l)Ks-S(l,\;n)\geq kKs\} \end{array}$$
Combined with the derivation of (Equation 15) and renting the stopping time theory for the martingale process $M_{l}^{L-s}\left(n-k\right)$, we can derive that
(20)$$\begin{array}{cc} \; & E\left[M_{l}^{L-s}\left(0\right)\right]=\prod_{i=1}^{2}E[V_{x_{i}\left(0\right)}^{a}]E\left[V_{s(0)}^{s}\right]\geq He^{\theta^{*}Ksk}P\left\{T_{W}\leq n\right\} \end{array}$$
When $T_{W}\to \infty$, we can obtain $P\left\{W\left(n\right)\geq k\right\}=P\left\{T_{W}\geq \infty \right\}$. The proof is completed.    □

## 4. Relay Selection for Visible Light Network

We assume that the visible light network is represented by G=(V,L). *V* is the set of all nodes in the network, and *L* is the set of all links in the network. The nodes are static and the communication links are bidirectional. The topology changes as nodes die or links fail. All visible light nodes have the same configuration.

We set the network operation by time sharing, i.e., $t\in \left\{0,1,2,\ldots \right\}$. At each time slot *t*, the back-pressure algorithm activates a set of interference-free links in the network. When the new data arrive in the network, each node makes routing and transmission scheduling decisions, and the visible light packets are delivered to the appropriate destinations. The back-pressure algorithm divides the visible light packet types according to the difference in the destination. Each packet in the network records information about the source and destination nodes. $Q_{i}^{d}\left(t\right)$ is denoted as the total number of packet types on node *i* at time slot *t*. If d=i, then $Q_{i}^{d}\left(t\right)=0$, indicating that node *i* is the destination node for the *d* number packets. Each node maintains up to *N* (total number of nodes) queues for storing packets arriving at different destination nodes.

π is denoted as the set of link scheduling, and Γ denotes the set of all schedulable links under the interference condition. The back-pressure algorithm used in the visible light network is as follows.

**STEP-1: Calculating the weights of the links.** The link (m,n) is any link in the network, the node *n* is a neighboring node of the node *m* within the network communication range, and *d* is the destination node in the network. We calculate the backlog difference of different kinds of packets between node *m* and node *n*, while the value of the backlog difference must be the positive integer, as follows,
(21)$$diff_{mn}^{d}\left(t\right)=\max\left\{Q_{m}^{d}\left(t\right)-Q_{n}^{d}\left(t\right),0\right\}$$
where $diff_{mn}^{d}\left(t\right)$ denotes the backlog difference of the packet queue of node *d* between node *m* and node *n* at time slot *t*. $Q_{m}^{d}\left(t\right)$ and $Q_{n}^{d}\left(t\right)$ are the queue length values for node *m* and node *n*, respectively.

We select the maximum value of all the queue backlog differences as the weight of the link, as follows,
(22)$$w_{mn}\left(t\right)=\max_{d\in D}\left\{diff_{mn}\left(t\right)\right\}$$
where *D* is the set of all destination nodes. $w_{mn}\left(t\right)$ is a maximum value in the set of all different kinds of packet queue backlog differences $diff_{mn}\left(t\right)$. At time slot *t*, we make $c_{mn}^{*}\left(t\right)$ as the optimal packet with $w_{mn}\left(t\right)$ value for link (m,n).

**STEP-2: Selecting schedulable links.** We set the optimization function as follows,
(23)$$\pi \left(t\right)=\underset{\pi \in \Gamma }{\arg\max}\sum_{m,n}\mu_{mn}\left(t\right)w_{mn}\left(t\right)$$
where Γ denotes the set of all schedulable links under the interference model. $\mu_{mn}$ is the link transmission rate. $\pi \left(t\right)$ is the set of optimal transmission links.

**STEP-3: Selecting routing paths.** At time slot *t*, link (m,n) under the routing and scheduling policy, we transmit the optimal kind of packet $c_{mn}^{*}\left(t\right)$ from node *m* to node *n*.

The back-pressure algorithm for the relay selection in the visible light network is shown in Algorithm 1. We illustrate the workflow of the algorithm with a simple example. Figure 2 shows a visible light communication uploading network. There are three relay nodes *C*, *D*, and *E*, and three different destinations *X*, *Y*, and *Z*. From the figure, we can see the backlog number, i.e., the queue length value, corresponding to each packet type at each node, as shown in Table 1. We assume that the link (C,D) rate is $\mu_{cd}=4$ packets/slot, and the link (C,E) rate is $\mu_{ce}=2$ packets/slot. The neighboring node *C* at time slot *t* are only *D* and *E*.
**Algorithm 1** The back-pressure algorithm for the relay selection in visible light network**Input:** G=(V,L), *m*, μ∖∖ Web *G*, Node *m*, Link rate μ**Output:** Routing strategy   1:**procedure** Original back-pressure▹Calculate Weight   2:**for** all links (m,n)∈L **do**   3:    **for** all d∈D**do**∖∖Packet type *d*   4:        $diff_{mn}^{d}\left(t\right)\leftarrow Q_{m}^{d}\left(t\right)-Q_{n}^{d}\left(t\right)$   5:    **end for**   6:    $w_{mn}\left(t\right)\leftarrow \max_{d\in D}\left\{diff_{mn}^{d}\left(t\right)\right\}$   7:    8:    $c_{mn}^{*}\left(t\right)\leftarrow \underset{d\in D}{\arg\max\{}diff_{mn}^{d}\left(t\right)\}$∖∖ Packet types with maximum weight   9:**end for**▹ Link Scheduling  10:**for** all π∈Γ **do**  11:    $sum_{\pi }\leftarrow \sum_{(m,n)\in L}\mu_{mn}\left(t\right)w_{mn}\left(t\right)$∖∖ The sum of the products of link weight and rate  12:**end for**  13:$\pi \left(t\right)\leftarrow \arg\max_{\pi \in \Gamma }sum_{\pi }$▹ Data Transferring  14:**for** all $(m,n)\in L:(Q_{m}^{d}\left(t\right)-Q_{n}^{d}\left(t\right))>0$ **do**  15:    transfer packets $c_{mn}^{*}\left(t\right)$ from *m* to *n*  16:**end for**  17:**end proceduce**

First, we use Equation (Equation 21) to compute the queue backlog difference for different kinds of packets *X*, *Y*, and *Z* for link (C,D) and link (C,E), respectively. The values of queue backlog difference for different kinds of packets are shown in Table 2.

Then, we calculate the weights of the two links (C,D) and (C,E), respectively, by Equation (Equation 22); the results are shown in Table 3.

Next, we derive the schedulable links. At time slot *t*, for node *C*, the link scheduling sets (C,D) and (C,E) interfere with each other, they cannot be scheduled simultaneously. The rate of link (C,D) is $\mu_{cd}=4$ packets/slot, the rate of link (C,E) is $\mu_{ce}=2$ packets/slot, and $\sum \mu_{CD}w_{CD}\left(t\right)=4\times 6=24$, $\sum \mu_{CE}w_{CE}\left(t\right)=2\times 6=12$. We can find that $\sum \mu_{CD}w_{CD}\left(t\right)>\sum \mu_{CE}w_{CE}\left(t\right)$, so we select link (C,D). The optimal data packet type is *Y* and the optimal transmission rate is $\mu_{CD}\left(t\right)$.

Finally, at time slot *t*, the routing decision of node *C* is that packets of type Y will be transmitted to node D at a rate of 4 packets/slot through link (C,D) for several packets.

## 5. Simulations and Results Analysis

Firstly, the derived theoretical result is evaluated in part A, and the relay selection algorithm is analyzed in part B.

***A.*** 
*
**The theoretical upper bound of the unreliability of latency**
*


The dynamic change of packets in the queuing system is simulated, where the arrival process models the aggregate traffic. The simulation results of the latency-bounded unreliability are measured by the ratio of the number of slots where the latency exceeds the threshold to the total observation slots.

In Figure 3, the aggregate arrival process is composed of an MMMP-4 process and an IMP process. The service process is modeled as a geometric distribution. The model parameters are summarized in Nomenclature section. The upper bounds of unreliability regarding latency with different system load ρ is presented, where ρ is defined as the ratio of the average arrival rate to the average service rate. According to the system load, the corresponding service rate *C* can be determined. The simulation results of unreliability are presented by box plots. It can be shown that the upper bounds of unreliability with regard to latency match the simulated results well with different ρ. The analysis framework can provide effective results of latency performance for the aggregate traffic with heterogeneity and burstiness. Obviously, the system load plays a significant role in the reliability. In the heavy-load system (ρ=0.9), the latency-bounded reliability is terrible such that it cannot support almost any services with statistical reliability requirement even if the latency threshold is large.

In order to verify the advantages of the proposed analysis framework, we compare the theoretical results with the state-of-the-art ones based on EB/EC theory in Figure 4. In this part, the aggregate traffic consists of two identical MMOO processes. The parameters of the MMOO model are listed in Nomenclature section. The result based on the EB/EC theory refer to [20]. It can be shown that the upper bound of unreliability obtained in this paper is much tighter than the compared one. The martingale-based theoretical results match the simulations well. The gap between the EB/EC-based results and the simulated values is obvious. Our analysis method can provide a more precise analysis for the latency performance of the Markovian aggregate traffic. The EB/EC theory assumes bursty traffic as the constant fluid, which makes it impossible to reflect the influence of complex random characteristics of aggregate traffic on latency performance.

Next, we analyze the impact of the burstiness of the aggregate traffic on the latency-bounded unreliability. Two measurement parameters of burstiness are adopted, which are the Hurst and squared coefficient of variation (CV). We change the state transition probabilities of the arrival models to influence the burstiness of aggregate traffic. The Hurst parameter is computed by the method of rescaled adjusted range analysis in this paper. Corresponding to the adjusted state transition probabilities ({0.1,0.01,0.001}), the Hurst parameter is {0.5904,0.8495,0.9140}, respectively. The CV of MMMP-4 can be derived as in (Equation 24), where $Rav_{b}$ is the average rate of the MMMP-4 model. A larger Hurst parameter or CV indicates a stronger burstiness of the traffic. The latency-bounded unreliability versus latency threshold with different Hurst parameters is shown in Figure 5, and the unreliability versus CV is in Figure 6. In Figure 5 and Figure 6, the system loads are 75% and 60%, respectively, which are relatively high. The latency performance decreases sharply with the burstiness. The system will become unsustainable when the system load is high along with a bursty arrival. With the CV rising from 5 to 20, the system could only provide a best-effort QoS in the current service configuration. For the aggregate traffic with complex stochastic features, the impacts of burstiness on the latency-bounded reliability must be considered.
(24)$$\begin{array}{c} CV=\frac{2Rav_{b}\left[{\left(p_{1}+q_{1}\right)}^{2}+\left(p_{1}\lambda_{1}R_{M1}+q_{1}\gamma_{1}R_{M2}\right)\left(1-p_{1}q_{1}\right)\right]}{\left(p_{1}+q_{1}\right)\left[q_{1}\lambda_{1}R_{M1}+p_{1}\gamma_{1}R_{M2}+\lambda_{1}\gamma_{1}R_{M1}R_{M2}\left(1-p_{1}-q_{1}\right)\right]}-Rav_{b}-1. \end{array}$$

Finally, the impacts of system load on the latency performance are analyzed. Figure 7 presents the results of latency-bounded unreliability versus Rav of the aggregate traffic, $Rav=Rav_{b}+Rav_{i}$. When Rav<4 packets/slot, the system load is lower than 66.7%. If the threshold *k* is relatively loose, the latency-bounded reliability is acceptable. However, for the business with strict latency threshold requirements (k=25 to 70), the unreliability increases linearly with the system load. System load is an essential factor impacting the latency performance. If the system load goes too high, latency performance is unsatisfactory even if the latency threshold is moderate and the average arrival is not too large. From another perspective, even if the load is not high, the strong burstiness of arrivals may cause latency thresholds (those relatively small) to be violated frequently. The dotted lines in Figure 7 support our analysis.

***B.*** 
*
**Relay selection algorithm performance**
*


We compare two relay selection algorithms: (1) our proposed relay selection algorithm based on the back-pressure theory, and (2) the traditional relay selection algorithm based on the max–min criterion method. In Figure 8, we show the evaluation of two relay selection algorithms through the average link throughput. From the figure, we can find that the average link throughput continues to increase with the increasing packet arrival rate, under two relay selection algorithms. Our proposed relay selection algorithm always achieves higher throughput than the traditional relay selection algorithm. It shows that the back-pressure theory can alleviate bandwidth limitations.

In Figure 9, we plot the relationship between the signal-to-noise ratio (SNR) threshold and outage traffic flows, under traditional and our proposed relay selection algorithm. If the SNR is still lower than the SNR threshold after the relay selection algorithm optimization, the transmission channel will be interrupted, i.e., outage traffic flows are interrupted. From the figure, we find that when the SNR threshold increases, both relay selection algorithms increase the number of outage traffic flows. However, the number of outage traffic flows from our proposed algorithm is significantly lower than that of the traditional algorithm. The result confirms that our proposed relay selection algorithm can provide better reliability network performance.

## 6. Conclusions

In this paper, we propose a novel latency-bounded reliability analysis framework based on martingales for the queuing system, where the aggregate traffic with burstiness is considered. The martingale of Markov-modulated arrival processes and the martingale of i.i.d. service processes are constructed, respectively. Leveraging the multiplicative property of martingales, the martingale of the backlog is proposed to model the buffer behavior of aggregate arrivals. Renting the stopping time theory, a tight upper bound of the unreliability with regard to latency for the aggregate traffic is derived. Meanwhile, in the core network, based on the back-pressure theory, a reasonable relay selection algorithm is proposed to boost the network performance and data transmission quality. Simulations verify the effectiveness of the proposed analysis framework. The latency performance is analyzed in terms of burstiness and the system load. When Hurst > 0.8, the latency-bounded unreliability is up to $10^{-1}$ level, and the system is unable to provision any reliability guarantee. For the aggregate traffic with sharp burstiness, more resources should be configured.

## Figures and Tables

**Figure 1 sensors-24-02748-f001:**
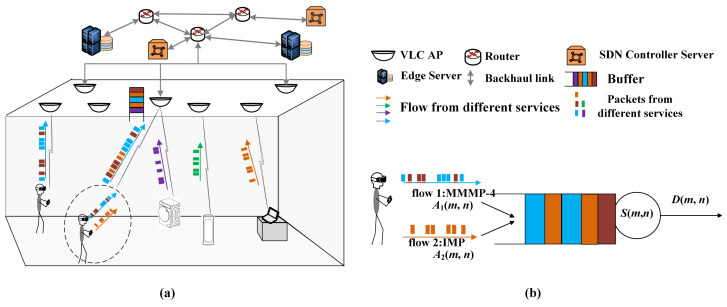
Scenario diagram. (**a**) is the network scenario. (**b**) is the corresponding queuing model.

**Figure 2 sensors-24-02748-f002:**
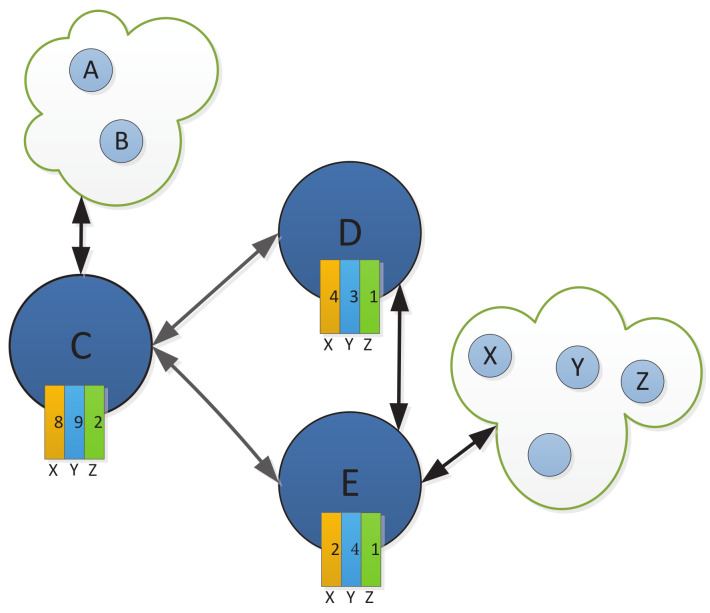
The visible light communication network for relay selection. Both A and B are source nodes, and there are three relay nodes, C, D, and E, and three different destinations, X, Y, and Z.

**Figure 3 sensors-24-02748-f003:**
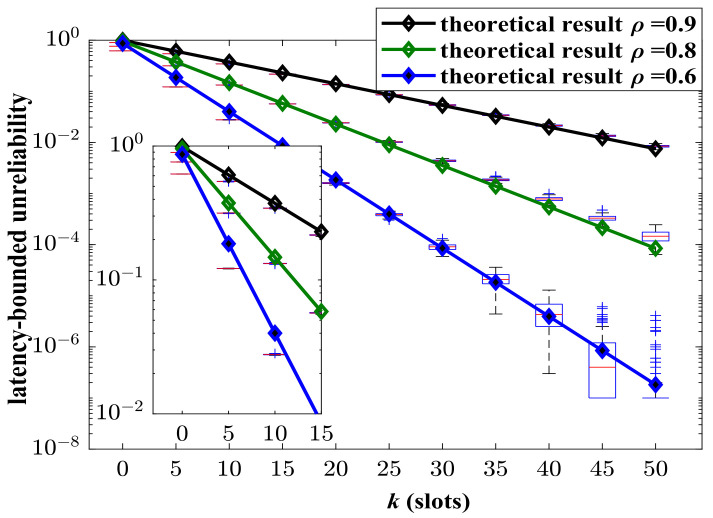
The simulated and theoretical results of latency-bounded unreliability, where the aggregate traffic is composed as a four-state Markov-modulated multinomial process (MMMP-4) and an interrupted multimomial process (IMP), $p_{1}=0.1$, $q_{1}=0.5$, $\lambda_{1}=0.4$, $\gamma_{1}=0.6$, $R_{M1}=3$ packets/slot, $R_{M2}=4$ packets/slot, $p_{2}=0.1$, $q_{2}=0.5$, $R_{I2}=20$ packets/slot, p=0.5.

**Figure 4 sensors-24-02748-f004:**
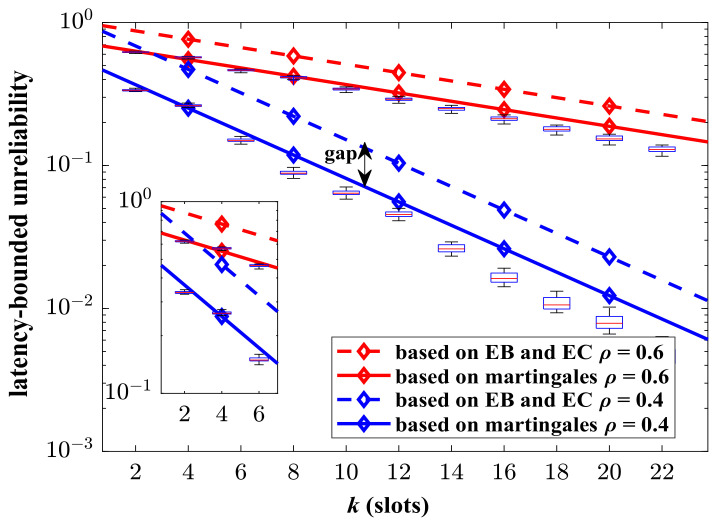
The simulated and theoretical results of latency-bounded unreliability, where the aggregate traffic is composed of two identical Markov modulated on–off (MMOO) processes, $p_{a}=0.1$, $q_{a}=0.5$, R=5 packets/slot, p=0.5.

**Figure 5 sensors-24-02748-f005:**
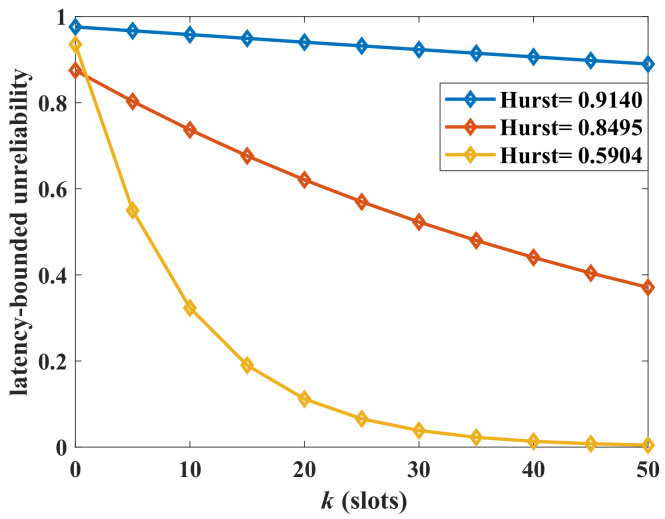
The latency-bounded unreliability of aggregate traffic with different Hurst parameters, where the aggregate traffic is composed as a four-state Markov-modulated multinomial process (MMMP-4) and an interrupted multimomial process (IMP), $p_{1}=q_{1}=p_{2}=q_{2}\in \left\{0.1,0.01,0.001\right\}$, $\lambda_{1}=0.4$, $\gamma_{1}=0.6$, $R_{M1}=30$ packets/slot, $R_{M2}=1$ packets/slot, $\gamma_{2}=0.6$, $R_{I2}=30$ packets/slot, p=0.5, C=40 packets/slot.

**Figure 6 sensors-24-02748-f006:**
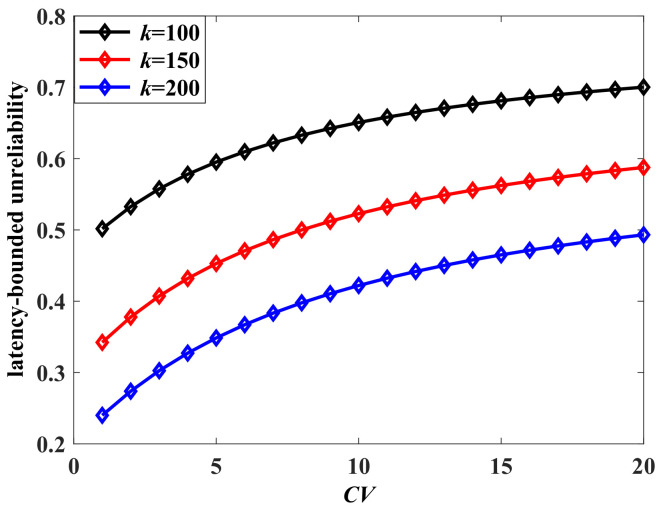
The latency-bounded unreliability of aggregate traffic vs. CV, where the aggregate traffic is composed as a four-state Markov-modulated multinomial process (MMMP-4) and an interrupted multimomial process (IMP), $p_{1}=0.1$, $q_{1}=0.5$, $\lambda_{1}=0.4$, $\gamma_{1}=0.6$, $Rav_{b}=5$ packets/slot, $p_{2}=0.1$, $q_{2}=0.5$, $\gamma_{2}=0.6$, $R_{I2}=10$ packets/slot, p=0.5, C=20 packets/slot.

**Figure 7 sensors-24-02748-f007:**
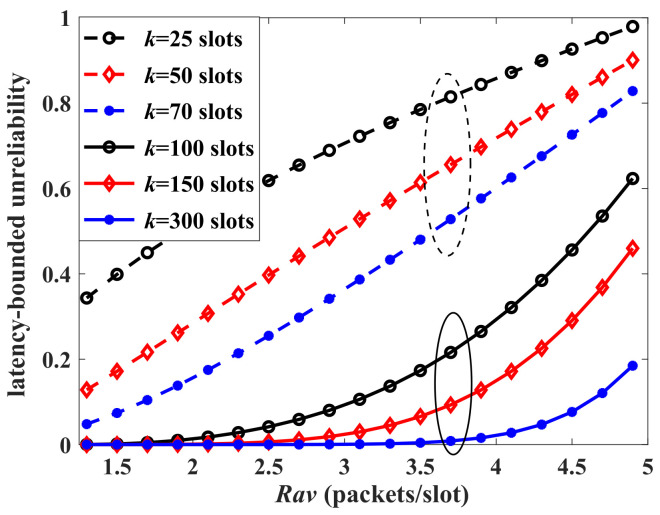
The latency-bounded unreliability of aggregate traffic vs. Rav, where the aggregate traffic is composed as a four-state Markov-modulated multinomial process (MMMP-4) and an interrupted multimomial process (IMP), $p_{1}=0.1$, $q_{1}=0.5$, $\lambda_{1}=0.4$, $\gamma_{1}=0.6$, $Rav_{b}\in \left[0.8,2.6\right]$ packets/slot, $p_{2}=0.1$, $q_{2}=0.5$, $\gamma_{2}=0.6$, for the IMP model, $Rav_{i}\in \left[0.5,2.3\right]$ packets/slot, p=0.5, C=12 packets/slot. The dashed circle corresponds to the strict threshold and the solid circle corresponds to the loose threshold.

**Figure 8 sensors-24-02748-f008:**
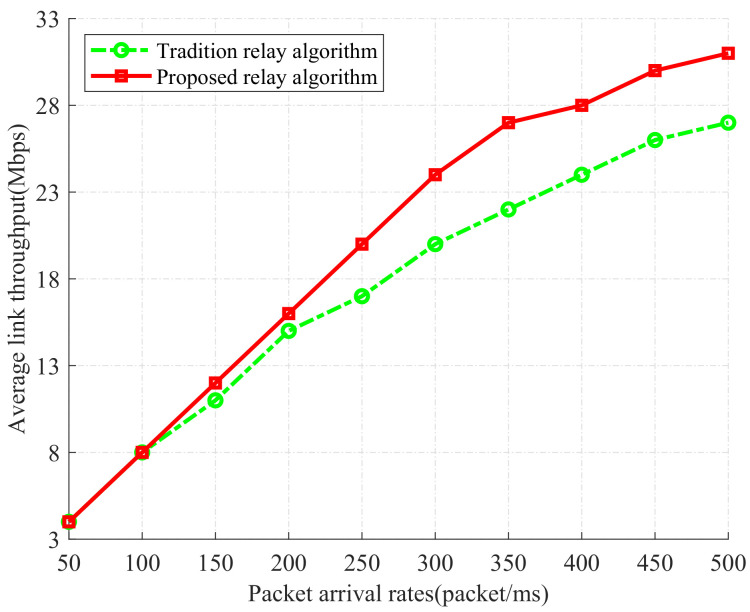
The relationship between packet arrival rates and link throughput under two different relay selection algorithms.

**Figure 9 sensors-24-02748-f009:**
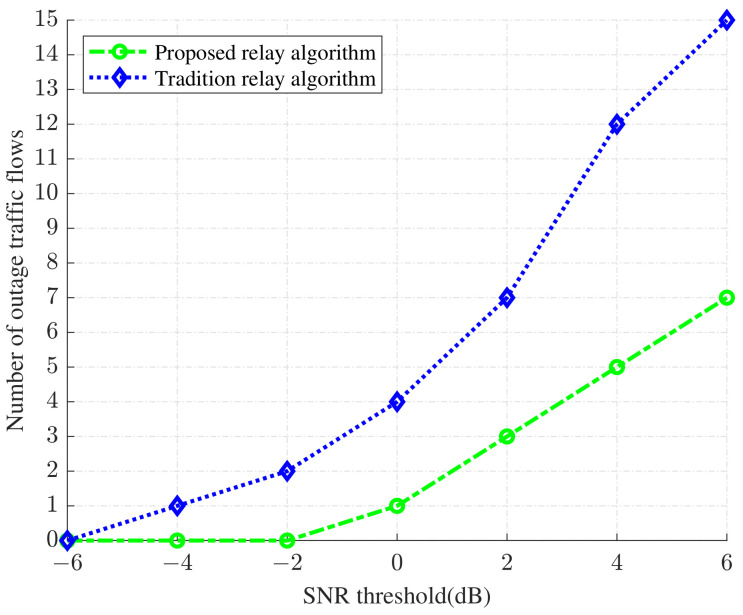
The relationship between signal-to-noise ratio (SNR) threshold and outage traffic flows under two different relay selection algorithms.

**Table 1 sensors-24-02748-t001:** The queue length in the node.

Length	Node	X	Y	Z
Typology				
C		8	9	2
D		4	3	1
E		2	4	1

**Table 2 sensors-24-02748-t002:** Different packet backlog values.

Backlog Difference	Link	X	Y	Z
Typology				
(C, D)		$diff_{CD}^{X}\left(t\right)=8-4=4$	$diff_{CD}^{Y}\left(t\right)=9-3=6$	$diff_{CD}^{Z}\left(t\right)=2-1=1$
(C, E)		$diff_{CE}^{X}\left(t\right)=8-2=6$	$diff_{CE}^{Y}\left(t\right)=9-4=5$	$diff_{CE}^{Z}\left(t\right)=2-1=1$

**Table 3 sensors-24-02748-t003:** Link weight.

Link	Weight
(C, D)	$w_{CD}\left(t\right)=\max_{d\in D}\left\{diff_{CD}\left(t\right)\right\}=6$
(C, E)	$w_{CE}\left(t\right)=\max_{d\in D}\left\{diff_{CE}\left(t\right)\right\}=6$

## Data Availability

The data presented in this study are available on request from the corresponding author.

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
