# Peer review of "The Latency Performance Analysis and Effective Relay Selection for Visible Light Networks"

_sensors, 2024, doi:10.3390/s24092748_

Round 1
Reviewer 1 Report
Comments and Suggestions for Authors
1. My main concern regarding the content of the article is the following: What exactly is the specificity of VLC in this case, why should the methods proposed in the article meet the features of VLC? In particular, an article from 2020 (link below) describes extremely similar models and methods without specifying exactly which applications and access networks they may be intended for. In considered article, you present a specific type of network - VLC, but do not specify what its features are and how they can affect the methods and models used for its analysis. Also, given that the considered article and the article from 2020 describe the same traffic model (MMMP-4 and IMP), and the same approach to Delay analysis (based on martingales), I would suggest removing from the considered article a detailed description of these models and methods and adding a link to article from 2020.
Yu, B., Chi, X. and Sun, H. (2020), Delay analysis for aggregate traffic based on martingales theory. IET Communications, 14: 760-767. https://doi.org/10.1049/iet-com.2019.0282
2. My other important concern relates to Section 4 and backpressure routing. This is a well-known routing algorithm, and it is not entirely clear what is innovative about its use in this case? Could you clarify and highlight this in the article, because at the moment I do not see what new is proposed. Also it would be better to refer to previous articles about backpressure routing in this Section.
3. I also have a few smaller comments:
3.1. There is an RF abbreviation in the beginning of the article, I suppose if would be better to use "Radio Frequency" instead of abbreviation.
3.2. Also, you mention "dilemma of RF networks in the next generation information technologies" and for my mind it is better either explain this dilemma in details, or reference the articles where this dilemma is explained.
3.3. In the Introduction you use the term "unascertainty of traffic". Could you please explain what does it mean?
3.4. In the section 2.A: "Service time intervals in the VLC AP are set to following a geometric distribution" Could you please explain why geometric distribution of service time has been chosen?
3.5. In the 2.B section: "the process of packet generation is random" What is the distribution of this random process? I suppose it is uniform distribution, but I think this must be clearly stated.
3.6. There is a small mistake in the Figure 5 -- in the node E, number of packets with destination Y should be 4, as well as in the Table 1.
3.7. Also, I suppose there is an mistake in the Algorithm 1 Step 4, the second arrow probably should be a minus.
3.8. Could you please highlight the simulated results in Figure 6? Currently they are barely visible.
3.9. At the bottom of page 13 should be link to the Table 4, not Table 1.
3.10. At the same place you claim that "Our analysis method is more precise". I'm not sure here, but it seems to me, it is better to focus on the fact that your method allows to determine the upper bound of unreliability.
3.11. At the page 14: "We change the parameters of the MMMP-4 model to influence the burstiness of aggregate traffic" It would be great to see exactly what parameters of the model you changed and how to change the Hurst parameter of the resulting traffic.
3.12. From Figures 8 and 9 it is not quite clear why even with the small burstiness (H = 0,5904, CV = 0) the unreliability is quite high (more than 0.1 for significant number of slots). Could you please check these Figures?
Reviewer 2 Report
Comments and Suggestions for Authors
The authors focus on the latency analysis based on combining the martingale of backlog and the time shift feature for a relay selection technique in VLC communications. Please find below several recommendations to improve the quality and readability of the manuscript:
1. The authors argued that VLC communications require latency reliability within the millisecond level to reach 99.99%-99.999%. However, based on the 3GPP requirements, the number of errors for URLLC should be at the scale of 'five 9s', i.e., 99.99999%. Please check and verify again based on reference [1] or 3GPP.
2. The transition from the VLC scenario to the queueing system requires changes in either Fig 1 or Fig 2 with several more explanations in the manuscript. The authors also do not provide some justification to support several assumptions, such as why the length of the data packet is fixed and why in FIFO pattern - please relate with VLC
3. At the beginning of Sect 3, please describe why Martingale-based research has been selected in the study—the reader may be unclear on this.
4. Sect 4—Please include a drawing to illustrate the relay deployment for the considered VLC environment. Alternatively, is it possible to improve Fig 1 to illustrate the need for a VLC relay in the study?
5. Fig. 6, what is the targetted level for latency-bounded unreliability based on potential VLC applications? Without establishing a reference or a benchmark, it is difficult for the readers to understand whether the MMMP+ IMP solution is able to meet the latency requirements.
6. The result from Fig. 6 also suggests that the solution only works at a moderate load (rho < 0.6)? If yes, please improve the discussion in Page 12/13
7. Please revise the conclusion to include relevant metrics/statistics from the significant results to support the function/performance of the proposed works.
Reviewer 3 Report
Comments and Suggestions for Authors
This paper investigates the latency performance of a VLC network and proposes the so-called martingale of backlog. Although many technical terns are mentioned, such as the stopping time theory, the back-pressure theory, the paper lacks clarity in its focus.
1. The authors should clearly articulate the key point of the paper, including the identified problem and the proposed solution. It is essential to emphasize the originality of the proposed scheme instead of simply combining the existing technical jargon such as stopping time theory and back-pressure theory.
2. The paper should explicitly outline the advantages of the proposed scheme compared to conventional approaches. In the simulation section, it's crucial to highlight the performance improvements achieved.
3. The inclusion of 22 references unrelated to VLC raises questions about the necessity of using VLC in the uplink and its relevance in this context. The authors should justify why VLC is chosen over other uplink approaches.
4. Clarification is needed regarding why the MMMP-4 and IMP flows are specifically considered in the multi-terminal uplink communication scenario. The authors should justify this choice or explain the exclusion of other types of flows.
5. Detailed descriptions of the uplink channel model, including the characteristics of LEDs and PDs, such as radiation pattern, transmit power, PD FoV, and PD responsivity, are necessary. Key communication parameters, such as data rate, should also be specified to provide a comprehensive understanding of the VLC channel model.
6. The paper should address whether there is interference between different branches of VLC signals and how indoor reflections influence the communication. These factors are crucial in evaluating the performance of the proposed scheme.
7. The paper must ensure that the theoretical framework encompasses the factors relevant to VLC communication. If the paper claims to be in the field of VLC networks, it should thoroughly consider the specifics of VLC communication scenarios to maintain relevance and credibility.
Comments on the Quality of English Languageno
Reviewer 4 Report
Comments and Suggestions for Authors
Similarity index is very high and also all the equations are copied from existing research work.
Less contribution and authors only implemented their algorithm for visible light communication.
There is no much discussion about related work and need of this research work
Comments on the Quality of English LanguageModerate English edition required
Round 2
Reviewer 1 Report
Comments and Suggestions for Authors
Thank you very much for your comments and clarification!
In most cases, your clarifications and changes in the article resolved my questions.
Some points still remain debatable for me. In particular, Section 4: to what extent can the application of a known routing method without noticeable changes in a new type of communication network have scientific novelty? But given that the rest of the article contains scientific novelty and generally meets the criteria of good scientific research, I believe that my doubt about Section 4 is not decisive.
Reviewer 3 Report
Comments and Suggestions for Authors
The authors have answered the questions well. No questions from my side.
Author Response
Thank you very much for your precious time and efforts expended in helping improve our paper. Thanks again for your positive comments on our work. The contributions of this manuscript have been highlighted. We have stated the motivation of our work more clearly and the innovation of the manuscript has been refined.
Reviewer 4 Report
Comments and Suggestions for Authors
Still the similarity index of the paper is high. its shows that authors have used existing model and equation and they claim that this is proposed model. I am not satisfy with authors responses.
Comments on the Quality of English LanguageNil
